# Molecular basis of sidekick-mediated cell-cell adhesion and specificity

Kerry M Goodman[1†], Masahito Yamagata[2,3†], Xiangshu Jin[1,4‡],
Seetha Mannepalli[1], Phinikoula S Katsamba[4,5], Göran Ahlsén[4,5],
Alina P Sergeeva[4,5], Barry Honig[1,4,5,6,7*], Joshua R Sanes[2,3*],
Lawrence Shapiro[1,5,7*]

[1]Department of Biochemistry and Molecular Biophysics, Columbia University, New York, United States; [2]Department of Molecular and Cellular Biology, Harvard University, Cambridge, United States; [3]Center for Brain Science, Harvard University, Cambridge, United States; [4]Howard Hughes Medical Institute, Columbia University, New York, United States; [5]Department of Systems Biology, Columbia University, New York, United States; [6]Department of Medicine, Columbia University, New York, United States; [7]Zuckerman Mind Brain and Behavior Institute, Columbia University, New York, United States

*For correspondence: bh6@cumc. columbia.edu (BH); sanesj@mcb. harvard.edu (JRS); shapiro@ convex.hhmi.columbia.edu (LS)

†These authors contributed equally to this work

Present address: ‡Department of Chemistry, Michigan State University, East Lansing, United States

Competing interests: The author declares that no competing interests exist.

**Abstract** Sidekick (Sdk) 1 and 2 are related immunoglobulin superfamily cell adhesion proteins required for appropriate synaptic connections between specific subtypes of retinal neurons. Sdks mediate cell-cell adhesion with homophilic specificity that underlies their neuronal targeting function. Here we report crystal structures of Sdk1 and Sdk2 ectodomain regions, revealing similar homodimers mediated by the four N-terminal immunoglobulin domains (Ig1–4), arranged in a horseshoe conformation. These Ig1–4 horseshoes interact in a novel back-to-back orientation in both homodimers through Ig1:Ig2, Ig1:Ig1 and Ig3:Ig4 interactions. Structure-guided mutagenesis results show that this canonical dimer is required for both Sdk-mediated cell aggregation (via *trans* interactions) and Sdk clustering in isolated cells (via *cis* interactions). Sdk1/Sdk2 recognition specificity is encoded across Ig1–4, with Ig1–2 conferring the majority of binding affinity and differential specificity. We suggest that competition between *cis* and *trans* interactions provides a novel mechanism to sharpen the specificity of cell-cell interactions.

## Introduction

In the vertebrate retina, light-sensitive photoreceptors synapse on interneurons; these interneurons process the information and pass it to retinal ganglion cells (RGCs), which send it to the brain (*Masland, 2012*). Highly stereotyped patterns of connectivity between the ~70 types of interneurons and ~30 types of RGCs render the latter sensitive to specific visual features such as motion or edges (*Sanes and Masland, 2015*). Synapses between these interneurons and RGCs form in the inner plexiform layer (IPL) of the retina, with arbors of each specific neuronal subtype confined to one, or a few, of the approximately 10 sublaminae (*Roska and Werblin, 2001*; *Sanes and Zipursky, 2010*).

Some aspects of this specific connectivity appear to be mediated by recognition molecules of the immunoglobulin superfamily (IgSF). Studies in chicks and mice have revealed that defined interneuron and RGC subtypes express one or more of 10 closely related IgSF members: Sdk1, Sdk2, Dscam, DscamL1, and Contactins 1–6 (CNTNs 1–6) in largely non-overlapping patterns (*Yamagata et al., 2002*; *Yamagata and Sanes, 2008*, *2012a*; *Fuerst et al., 2008*, *2009*; *Shekhar et al., 2016*). In chick, Sdk, Dscam and CNTN family proteins are present as interneuron-RGC synapses form, and both knockdown and over-expression experiments show that they are necessary and sufficient for

directing neural processes to particular sublaminae in the IPL (*Yamagata et al., 2002*; *Yamagata and Sanes, 2008*, *2012a*). In mice, Sdk1, Sdk2, Dscam, DscamL1 and CNTN5 mutants each exhibit specific defects in arborization and connectivity within the IPL (*Fuerst et al., 2008*, *2009*; *Krishnaswamy et al., 2015*; Peng et al., unpublished). In one case, the specific connectivity of an interneuron type (vesicular glutamate transporter 3-positive amacrine cells or VG3-ACs) to a specific RGC type (W3B-RGCs) depends upon expression of Sdk2 in both cell types: transmission from VG3-ACs to W3B-RGCs fails in Sdk2 mutants and the RGCs no longer respond to their canonical visual feature (*Krishnaswamy et al., 2015*). These results have led to the hypothesis that IgSF-mediated homophilic interactions bias synaptic connectivity in favor of appropriate partners, thus generating information processing circuits in the retina. Since all 10 of these IgSF molecules are also expressed by neuronal subsets throughout the central nervous system (*Yamakawa et al., 1998*; *Agarwala et al., 2001*; *Shimoda and Watanabe, 2009*; *Stoeckli, 2010*; *Yamagata and Sanes, 2012a*), similar interactions may mediate connectivity in multiple brain regions.

Sdk1 has also been shown to be involved in the pathology of focal segmental glomerulosclerosis and HIV-associated neuropathy (*Kaufman et al., 2004*, *2007*, *2010*). Inappropriate up-regulation of Sdk1 expression by podocytes has been linked to their dedifferentiation and loss of proper foot-process architecture, leading to collapsed glomeruli and neuropathy (*Kaufman et al., 2007*). Sdk1 is normally expressed at high levels during kidney development, with very low expression afterwards. Sdk1-associated kidney pathologies are thought to reflect a reversion of podocytes to the early developmental state, caused by inappropriate Sdk1 expression (*Kaufman et al., 2004*, *2007*, *2010*).

Sdk1 and Sdk2 are single-pass transmembrane proteins, with extracellular regions composed of 6 N-terminal immunoglobulin (Ig) domains followed by 13 fibronectin type III (FNIII) domains, and a relatively short intracellular domain terminating in a Postsynaptic density/Discs Large/ZO-1 (PDZ) binding motif (*Figure 1A*) (*Nguyen et al., 1997*; *Yamagata et al., 2002*; *Yamagata and Sanes, 2010*; *Kaufman et al., 2010*). Binding of this C-terminal motif to scaffolding molecules of the membrane-associated guanylate kinase with inverted orientation (MAGI) family is necessary for synaptic localization of Sdks, and required for appropriate function in the retina and kidney (*Yamagata and Sanes, 2010*; *Kaufman et al., 2010*).

IgSF neural recognition proteins from the Dscam, CNTN and L1 families have an extracellular domain architecture related to the Sdks: Dscam and DscamL1 have 10 extracellular Ig and 6 FNIII domains (*Schmucker and Chen, 2009*); CNTNs contain 6 Ig and 4 FNIII domains (*Shimoda and Watanabe, 2009*); and L1-related molecules have 6 Ig and 4 or 5 FNIII domains (*Maness and Schachner, 2007*). The four N-terminal Ig domains are arranged in a horseshoe conformation in

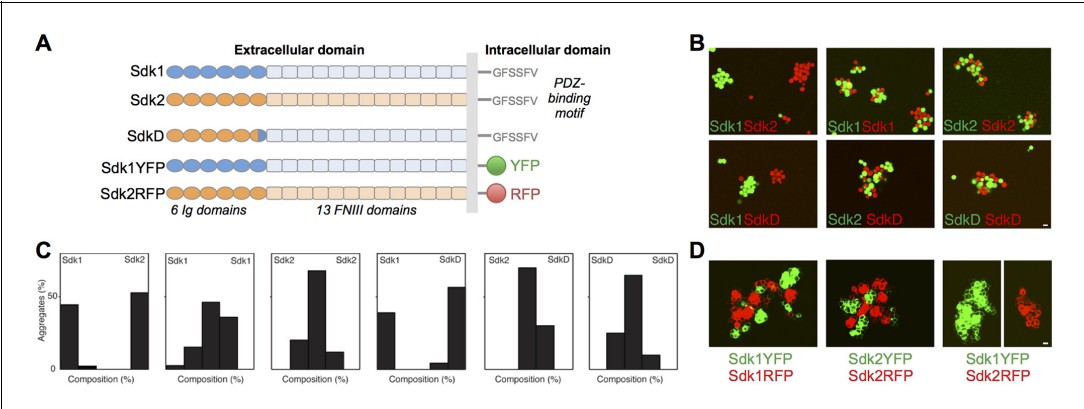

**Figure 1.** The Sdks' Ig domain regions are responsible for Sdk1/2 specificity. (**A**) Schematic of the domain arrangement of mouse Sdk1, Sdk2, and Sdk constructs used in the cell assays. (**B**) Aggregation assays demonstrate that L cells (labeled with red or green cell-trackers) co-aggregate only when the cells are expressing the same Sdk (top row). Cells expressing the SdkD chimera (shown in **A**) co-aggregate with Sdk2 but not Sdk1 expressing cells (bottom row). Scale bar, 5 μm. (**C**) Quantification of the assays shown in **B**. Each graph shows the proportion of red and green cells observed in the cell aggregates in each assay (n = 20–39). (**D**) HEK-293 cells transiently transfected with Sdk-RFP and YFP constructs show cell surface fluorescence and only co-aggregate with cells expressing the same Sdk molecule, regardless of the intracellular fluorescent tag. Scale bar, 5 μm.

Dscams, CNTNs, and L1-family proteins (*Chen et al., 2013*). Despite this similarity in protomer architecture, crystallographic studies of the *Drosophila* Dscam ortholog, Dscam1 (*Meijers et al., 2007*; *Sawaya et al., 2008*), human CNTN2 (Axonin-1/TAG-1) (*Mörtl et al., 2007*), mouse CNTN4 (*Bouyain and Watkins, 2010*), and the human L1 family member Neurofascin (*Liu et al., 2011*), revealed distinct homodimer structures mediated by horseshoe motifs.

Here, we report the crystal structures of cell-cell adhesive homophilic dimers of mouse Sdk1 and Sdk2, each mediated by the four N-terminal Ig domains. These four domains adopt a horseshoe conformation, like many other IgSF cell-cell recognition proteins, but they interact in a unique back-to-back anti-parallel manner not previously observed. Mutagenesis studies both in vitro, with analytical ultracentrifugation (AUC) and surface plasmon resonance (SPR) readouts, and in situ with a cell aggregation assay readout, demonstrate that the crystallographic dimer is present in solution and is required for Sdk-mediated cell aggregation. Interestingly, this same dimer is also required for *cis*-clustering of Sdk-molecules in isolated cells. Structures of multiple crystal forms of Sdk1 and Sdk2 revealed an unexpected flexibility in the dimer arrangement: The antiparallel contacts between Ig1–2 regions from each protomer were maintained in all structures, however the degree of contact between the Ig3–4 regions was highly variable. Consistent with this observation, mutagenesis studies showed that the Ig1–2:Ig1–2 interaction was necessary for dimerization, whereas mutations that interfered with the Ig3–4:Ig3–4 interaction had only a modest effect on dimer affinity. Overall, our data suggest a model in which Sdks form *cis* dimers on isolated cell surfaces, which dissociate to form *trans* dimers through the same interface when contact is made to a cell surface expressing the cognate Sdk. Competition between these *cis* and *trans* dimers may provide a mechanism to enhance the homophilic specificity of Sdk-mediated interactions.

## Results

### The adhesive Sidekick dimer is mediated by Ig1–4

Consistent with their role in defining neuronal contacts, both Sdk1 and Sdk2 mediate homophilic adhesion when applied to beads or transfected into cultured cells (*Yamagata et al., 2002*; *Yamagata and Sanes, 2008*; *Figure 1*). A chimeric construct (SdkD, *Figure 1A*) comprising Ig1–5 and part of Ig6 from Sdk2 and the remainder of the molecule from Sdk1 could mediate adhesion to Sdk2 but not Sdk1 in a mixed cell aggregation assay, using either L cells (*Figure 1B and C*) or N-cadherin deficient HEK-293 cells (data not shown), indicating that it is the Ig domain region that mediates cell-cell recognition in common with other IgSF proteins (*Gouveia et al., 2008*; *Haspel et al., 2000*; *Liu et al., 2011*; *Wojtowicz et al., 2004*; *Sawaya et al., 2008*). We also asked whether the cytoplasmic domain is required for cell-cell adhesion. To this end, we replaced the cytoplasmic domains of Sdk1 and Sdk2 with fluorescent proteins. Adhesion was unperturbed by this replacement (*Figure 1D*). Thus Sdk-mediated cell-cell adhesion requires the extracellular but not the intracellular domains of the proteins, with key determinants of homophilic specificity in Ig1–6.

To further define and measure the adhesive interaction for mouse Sdk1 and Sdk2, we produced soluble Ig1–4, Ig1–5 and Ig1–6 constructs in HEK-293 cells. Sedimentation equilibrium analytical ultracentrifugation (AUC) measurements showed that Sdk1 and Sdk2 Ig1–4, Ig1–5, and Ig1–6 were each dimers in solution with low-micromolar affinities (*Table 1*) with the Sdk2 dimer exhibiting ~5-fold stronger affinity than the Sdk1 dimer for each truncation construct tested. These affinities are similar to other cell-cell recognition proteins, such as *Drosophila* Dscam1 isoforms (1–2 μM; *Wu et al., 2012*) and classical cadherins (8–130 μM; *Harrison et al., 2011*; *Vendome et al., 2014*). Ig1–4 is therefore sufficient for dimerization in solution for both Sdks. We further note that the Ig1–6 constructs for both Sdk1 and Sdk2 gave 4–5-fold stronger dimerization affinities than the Ig1–4 constructs (*Table 1*), However, the addition or deletion of domains that do not participate in the interface frequently lead to small changes in binding energy, and this does not always reflect the presence of additional interactions. For example, we previously observed human VE-cadherin EC1–5 to have ~4-fold stronger dimerization affinity than the EC1–2 fragment (1.03 vs. 4.38 μM), even though the entire dimerization interface is contained within EC1 (*Brasch et al., 2011*). The effect of additional domains on the binding affinities may be due in part to entropic differences in the unbound state whereby crowding effects may affect the conformational freedom for the longer constructs.

**Table 1.** Sedimentation equilibrium analytical ultracentrifugation of Sdk fragments. $K_D$ = dissociation constant. $K_I$ = isodesmic constant. The $K_I/K_D$ ratio is given when it is less than two, indicating the presence of non-specific binding.

| Protein | Oligomeric state | Dimerization $K_D$ (µM, n=2) |
| --- | --- | --- |
| **Sdk1** | | |
| Ig1–4 | Dimer | 10.5 ± 1.1 |
| Ig1–5 | Dimer | 4.6 ± 0.06 |
| Ig1–6 | Dimer | 2.3 ± 0.39 |
| *Ig1–2 dimer interface mutations* | | |
| Ig1–4 N22R | Monomer | N/A |
| Ig1–4 K133E | Weak non-specific dimer | 204 ± 38.9 ($K_I/K_D$ = 1.24) |
| Ig1–4 L29M/E168D | Dimer | 4.26 ± 0.50 |
| Ig1–6 N22R | Monomer | 650 ± 66 ($K_I/K_D$ = 1.05) |
| *Ig3–4 dimer interface mutation* | | |
| Ig1–4 N253E | Dimer | 15.5 ± 1.64 |
| **Sdk2** | | |
| Ig1–4 | Dimer | 2.2 ± 0.4 |
| Ig1–5 | Dimer | 0.73 ± 0.036 |
| Ig1–6 | Dimer | 0.44 ± 0.012 |
| *Ig1–2 dimer interface mutations* | | |
| Ig1–4 H18R/N22S | Monomer | N/A |
| Ig1–4 N22S | Monomer | N/A |
| Ig1–4 N22R | Monomer | N/A |
| Ig1–6 H18R/N22S | Monomer | N/A |
| *Ig3–4 dimer interface mutation* | | |
| Ig1–4 N253E | Dimer | 18.9 ± 0.95 |
| **Chimera** | | |
| Sdk2$_{Ig1–2}$/Sdk1$_{Ig3–4}$ | Dimer | 3.92 ± 0.17 |

## The four N-terminal Ig domains are arranged in a stable horseshoe conformation

To determine the nature of the adhesive interaction for both Sdk1 and Sdk2, we determined the crystal structures of Sdk1$_{Ig1–4}$, Sdk1$_{Ig1–5}$, Sdk2$_{Ig1–4}$, and an Sdk2$_{Ig1–2}$/Sdk1$_{Ig3–4}$ chimera (*Figure 2A*). Two crystal forms of Sdk1$_{Ig1–4}$ were determined at 2.2 and 3.2 Å resolution respectively, one of Sdk1$_{Ig1–5}$ at 3.5 Å, two of Sdk2$_{Ig1–4}$ at 2.7 and 3.2 Å respectively, and one of the Sdk2$_{Ig1–2}$/Sdk1$_{Ig3–4}$ chimera at 2.7 Å resolution. Data collection and refinement statistics are given in *Figure 2—source data 1*.

The N-terminal four Ig domains of both Sdk1 and Sdk2, and the chimeric construct, Sdk2$_{Ig1–2}$/Sdk1$_{Ig3–4}$, are arranged in highly similar horseshoe structures (*Figure 2A*), with pairwise root mean square deviations over aligned Cα atoms (RMSDs) among all protomers of 2.4 Å or less.

The horseshoe conformation is formed by an anti-parallel interaction between Ig1–2 and Ig3–4, which are connected by an eight amino acid linker between Ig2 and Ig3 (Sdk1 R185–A192 and Sdk2 N185–P192). Two non-overlapping interfaces hold the horseshoe together: a conserved and relatively rigid Ig1:Ig4 interface and a more flexible and varied Ig2:Ig3 interface (*Figures 2B–D* and *Supplementary file 1A*). These intramolecular interactions bury extensive combined intramolecular surface areas of 2620 Å$^2$ in Sdk1, 2459 Å$^2$ in Sdk2 and 2567 Å$^2$ in the Sdk2$_{Ig1–2}$/Sdk1$_{Ig3–4}$ chimera, implying that the horseshoe should be a stable element in all of these molecules. In this conformation, Ig1 and Ig2 are arranged in tandem with a ~40° bend between them; Ig3 and Ig4 are also

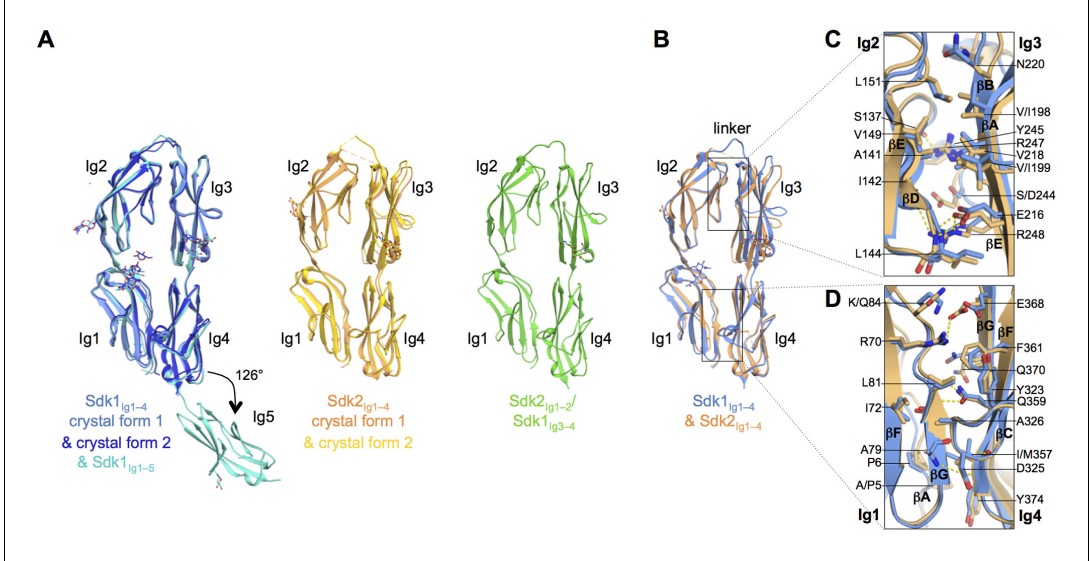

**Figure 2.** The N-terminal four Ig domains of Sdk1 and Sdk2 adopt a stable horseshoe conformation. (A) Crystal structures of the five or four N-terminal Ig domains of Sdk1 and Sdk2 show a horseshoe arrangement of Ig1–4, which is maintained in a Sdk2$_{Ig1–2}$/Sdk1$_{Ig3–4}$ chimera, from which Ig5 extends at a 126° angle from Ig4 in Sdk1. The left hand panel shows a structural alignment of single protomers from the two different Sdk1$_{Ig1–4}$ crystal forms and the Sdk1$_{Ig1–5}$ structure, showing the very high structural similarity of individual protomers among the different crystal forms. The second panel shows protomers from the two Sdk2$_{Ig1–4}$ crystal forms structurally aligned, and the third panel shows a single protomer from the Sdk2$_{Ig1–2}$/ Sdk1$_{Ig3–4}$ structure. All protein structures are shown in ribbon depiction, with oligosaccharides in stick view. (B) Structural alignment of the Sdk1 and Sdk2 Ig1–4 structures revealing that their overall architecture is highly similar. (C) Close-up of the Sdk1 (blue) and Sdk2 (orange) Ig2:Ig3 intra-horseshoe interfaces with interfacial residue side-chains and key main-chain atoms shown as sticks. Sdk2 hydrogen bonds are depicted as dashed yellow lines. (D) Close-up of the Ig1:Ig4 intra-horseshoe interface.

The following source data and figure supplement are available for figure 2:

**Source data 1.** X-ray crystallography data collection and refinement statistics.

**Figure supplement 1.** IgSF proteins containing horseshoe motifs including both Sdks have conserved intramolecular Ig1:Ig4 interactions.

arranged in tandem with a ~30° inter-domain bend (*Supplementary file 1A*). Horseshoe conformations are also found for the N-terminal four Ig domains of a number of other IgSF molecules including all members of the deleted in colorectal cancer (DCC), CNTN, L1, Dscam, cell adhesion associated oncogene regulated (CDON), MAM domain–containing glycosylphosphatidylinositol anchors (MDGA), and activated leukocyte cell adhesion molecule (ALCAM) families (*Chen et al., 2013*). Crystal structures of the horseshoe regions are available for *Drosophila* Dscam1 (*Meijers et al., 2007*; *Sawaya et al., 2008*) and vertebrate CNTN (*Freigang et al., 2000*; *Mörtl et al., 2007*; *Bouyain and Watkins, 2010*), DCC (*Chen et al., 2013*), and L1 family members (*Liu et al., 2010*; *Su et al., 1998*). Comparison between each of these structures and Sdk reveals the horseshoe conformation is similar overall (RMSDs from 2.8 to 6.2 Å; *Supplementary file 1B*) despite sequence identities with Sdk1 of only 20–26%.

The internal Ig1:Ig4 interface in both Sdk1 and Sdk2 includes conserved 'horseshoe motif' hydrogen bonds, identified by *Chen et al. (2013)* from their analysis of a number of horseshoe proteins. These occur between the side chains of Ig4 residues D325 (from Ig4 C strand WXXN/D motif) and Q359 (from Ig4 F strand ΦY/F/LQC motif) and the main chain of Ig1 residues A79 and L81 (*Figure 2D* and *Figure 2—figure supplement 1*). Despite the presence of these motifs and the requisite long Ig2–Ig3 linker, Chen et al., who surveyed the presence of IgSF horseshoe motifs through sequence analysis, did not identify Sdks as horseshoe motif-containing molecules. This is because Sdks' lack an internal disulfide in Ig2, which Chen et al. used to define the Ig2–Ig3 linker length in their bioinformatics search. The internal Ig2:Ig3 interface appears to lack conserved features amongst IgSF horseshoe proteins (*Chen et al., 2013*). In Sdk1 and Sdk2 the Ig2:Ig3 interaction is

mediated by the main chain to side chain hydrogen bonds between Sdk-conserved residues (S137$_{Ig2}$:R247$_{Ig3}$, I142$_{Ig2}$:R248$_{Ig3}$, and L144$_{Ig2}$:E216$_{Ig3}$), alongside van-der-Waals interactions among hydrophobic residues (L144$_{Ig2}$, V149$_{Ig2}$, L151$_{Ig2}$, V/I198$_{Ig3}$, V/I199$_{Ig3}$, and V218$_{Ig3}$) (*Figure 2C*). Remarkably, the Sdk2$_{Ig1-2}$/Sdk1$_{Ig3-4}$ chimera structure features the same set of hydrogen bonds and hydrophobic interactions that mediate both Ig1:Ig4 and Ig2:Ig3 interfaces.

The Sdk1$_{Ig1-5}$ structure showed that Ig5 extends laterally from the horseshoe with a 54° deviation from linearity with Ig4 (*Figure 2A*). This angle is maintained in both independent Ig1–5 chains observed in the crystal structure. The linker between Ig4 and Ig5 is only one amino acid (N379) and there is a small Ig4:Ig5 interface with a buried surface area (BSA) over both domains of ~370 Å$^2$. This interface—involving the linker region, the Ig4 AB loop and the Ig5 BC loop—likely provides rigidity to the Ig4–Ig5 junction. In Sdk2 the Ig4–Ig5 linker is the same length as in Sdk1, although the linker residue is serine rather than asparagine, and the Ig4:Ig5 interfacial residues are mostly conserved between Sdk1 and Sdk2 (*Figure 4—figure supplement 1*), except key Ig4 AB loop residue 302 which is a methionine in Sdk2 rather than a valine. The arrangement of Ig4 and Ig5 in Sdk2 is therefore likely to be similar to that observed in Sdk1. However the surface of the Ig4–Ig5 linker region in Sdk1 is highly acidic, whilst that of Sdk2 is comparably neutral (*Figure 4—figure supplement 2*).

We were unable to obtain crystals of an Sdk1 or Sdk2 construct containing Ig6 that diffracted to sufficient resolution for structure determination. The Ig5–Ig6 linker is ~2–4 amino acids long for both Sdk1 and Sdk2 (*Figure 4—figure supplement 1*), which could potentially accommodate a range of Ig5–Ig6 bend angles. The positioning of Ig6 is therefore unknown, however, since Ig5 projects away from the dimer interface it is unlikely Ig6 would be able to contribute to the dimer observed in the crystal structures, which is described in detail below.

## The Sidekick dimer is mediated by a flexible back-to-back interaction between horseshoes

Consistent with the AUC results of the wild-type Sdk1 and Sdk2, the crystal structures show a dimeric arrangement of molecules, consisting of symmetrical back-to-back interactions (convex-face to convex-face) between the Ig1–4 horseshoe regions of the individual protomers, with Ig5 making no contacts in the Sdk1$_{Ig1-5}$ dimer (*Figure 3A*). The dimer protomers are related by crystallographic 2-fold symmetry in both of the Sdk1$_{Ig1-4}$ structures, and by non-crystallographic symmetry in all the other structures. The Sdk1 and Sdk2 dimers are predominantly mediated by symmetrical anti-parallel interactions between the Ig1–2 halves of the horseshoes, with a contribution from an anti-parallel interaction between the Ig3–4 halves of the horseshoes observed in some, but not all of the crystal forms (*Figures 3B and 3C*).

The Ig1–2:Ig1–2 interactions in both the Sdk1 and Sdk2 structures consist of an Ig1:Ig1 interface near the dimer two-fold axis, and two symmetry-related Ig1:Ig2 interfaces. The Ig1:Ig1 interface, for both Sdk1 and Sdk2, is centered on N22, which hydrogen bonds with the main chain of R23, and is supported by interactions between hydrophobic residues (L19, V25) (*Figures 4A and 4B*). The Sdk1 and Sdk2 Ig1:Ig2 interfaces, which are contiguous with the central Ig1:Ig1 interface, consist of two networks of hydrogen bonding interactions clustered around residues E31$_{Ig1}$ and E/D168$_{Ig2}$; a salt bridge between E31$_{Ig1}$ and K133$_{Ig2}$ (corresponding to a region of complementary electrostatic potentials between Ig1 and Ig2, *Figure 4—figure supplement 2*); and a number of hydrophobic residue contacts including V4$_{Ig1}$:I/P135$_{Ig2}$ and L/M29$_{Ig1}$:V166$_{Ig2}$ (*Figures 4A and 4B*). The Ig1:Ig1 and Ig1:Ig2 interfaces of Sdk1 and Sdk2 are remarkably similar, with only five relatively conservative differences in the identity of the residues involved: I/V17$_{Ig1}$ (marginally interfacial), L/M29$_{Ig1}$, K/R55$_{Ig1}$, I/P135$_{Ig2}$, and E/D168$_{Ig2}$ (*Figures 4A–C* and *Figure 4—figure supplement 1*). There are also conserved differences in Ig1 residues 13–15 (GLP in Sdk1 and VRT in Sdk2; *Figure 4—figure supplement 1*), which form a loop near, but not within, the Ig1:Ig1 interface. The role of interacting variable residues 29$_{Ig1}$ and 168$_{Ig2}$ in Sdk1/Sdk2 specificity is discussed below.

Although the Ig1–2:Ig1–2 interaction is present in all of the crystal forms of Sdk1 and Sdk2, there is a difference in rotational angle between the two horseshoes of the dimer among the crystal forms, with the Ig1–2:Ig1–2 interface acting as the hinge (*Figures 3B and C*). This results in differing levels of interaction between the Ig3–4 regions in the Sdk1 and Sdk2 dimers in the different crystal forms (*Figure 3C*). The Ig3–4:Ig3–4 interaction varies from the Ig3–4 domains forming a considerable additional interface—as in Sdk1$_{Ig1-4}$ crystal form 2 (915 Å$^2$ BSA) and Sdk2$_{Ig1-4}$ crystal form 1 (1294 Å$^2$ BSA)—to being splayed 14 Å apart, as in the Sdk1$_{Ig1-4}$ crystal form 1 structure (0 Å$^2$ BSA). These

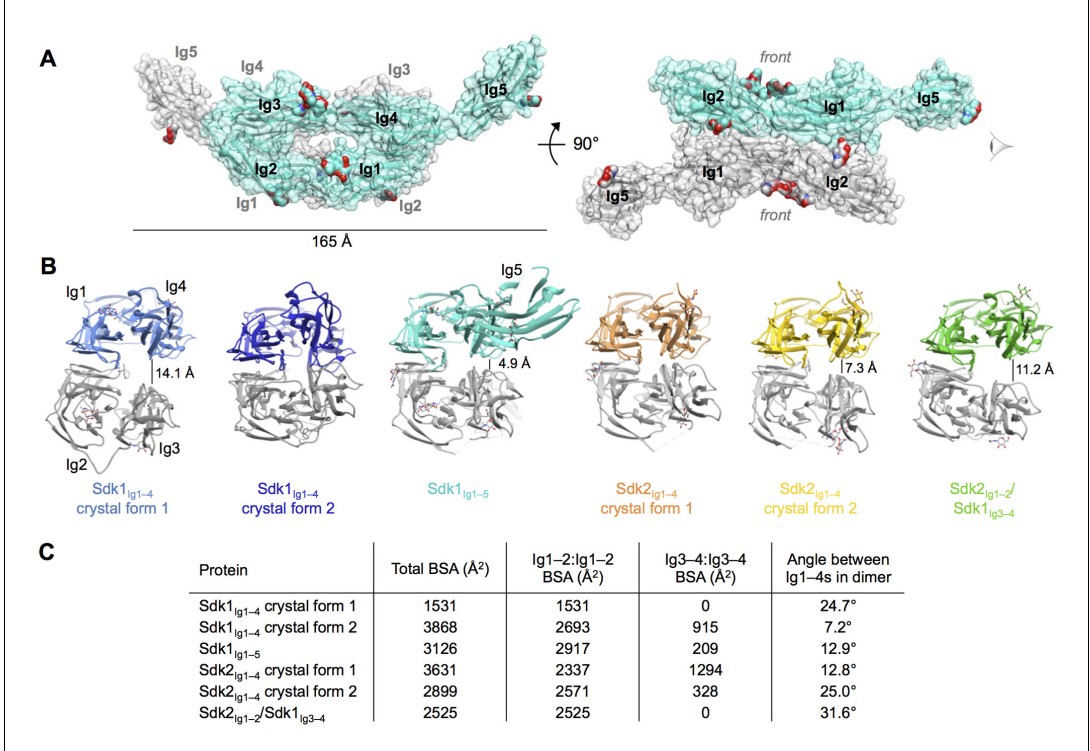

**Figure 3.** Crystal structures of Sdk1 and Sdk2 reveal a flexible dimer of horseshoes mediated by the N-terminal four Ig domains. (**A**) Surface view of the Sdk1$_{Ig1-5}$ structure showing the back-to-back horseshoe-mediated dimer observed in the crystal structure. One protomer is colored cyan, one protomer colored gray. Oligosaccharides are shown as opaque surfaces, colored by element. (**B**) The same back-to-back dimer is observed in all Sdk structures, but the angle between the two protomers in each dimer varies. Dimer structures are shown in ribbon depiction, viewed from the eye icon in **A**. In each dimer, one protomer is colored variably and one protomer is colored gray. All variably colored protomers are aligned, so that the difference in angle is evident from the differing positions of the gray protomer. (**C**) Table detailing the variation amongst the Sdk dimer crystal structures in the buried surface area (BSA) over both protomers, and in the angle between the planes of the two horseshoe regions (Ig1–4) in the dimers.

The following figure supplement is available for figure 3:

**Figure supplement 1.** Dimer interactions mediated by IgSF horseshoe motif proteins.

differences indicate that there is flexibility in the interaction between the horseshoes in solution, with different conformations being trapped in the crystals. These observations raise questions about how much the Ig3–4:Ig3–4 interfaces contribute to dimer stability and whether in cells this flexibility allows the accommodation of an as yet unidentified molecule in the Ig3–4:Ig3–4 clefts.

Analysis of the Ig3–4:Ig3–4 interface regions in the Sdk1$_{Ig1-4}$ crystal form 2 and Sdk2$_{Ig1-4}$ crystal form 1 structures, which show the most extensive Ig3–4:Ig3–4 contacts, reveals predominantly hydrophilic surfaces with hydrogen bonds between N253$_{Ig3}$ and P287$_{Ig4}$ in both structures, S240$_{Ig3}$ and T291$_{Ig4}$ in Sdk1, and T255$_{Ig3}$ and E286$_{Ig4}$ in Sdk2 (**Figures 4A and B**). The Ig3–4:Ig3–4 interface is not as highly conserved as the Ig1–2:Ig1–2 interface even within Sdk1 and Sdk2 homologs (**Figure 4C** and **Figure 4—figure supplement 1**). However, there are three Ig3–4:Ig3–4 interfacial residues that show non-conservative differences between Sdk1 and Sdk2 that are mostly conserved amongst vertebrate orthologs (H/S243$_{Ig3}$, Y/Q289$_{Ig4}$ and S/R296$_{Ig4}$) (**Figure 4C**). These differences result in the Sdk2 interface containing a π-stacking interaction between Y245$_{Ig3}$ and R296$_{Ig4}$, which is absent from Sdk1 (**Figures 4A and B**). Similarly, the Sdk1 interface contains an H243$_{Ig3}$:E293$_{Ig4}$ interaction, which is absent from Sdk2, although the contribution of this interaction is likely small since the side chains are still 4.4 Å apart in the Sdk1 crystal structure with the most extensive Ig3–4:Ig3–4 interface (**Figure 4A**).

In addition to these major interfacial regions, the Sdk1$_{Ig1-4}$ crystal form 2 also shows a small contact region between Ig1 residues E11 and R83 and Ig3 residue E212 (**Figure 4A**) towards the center

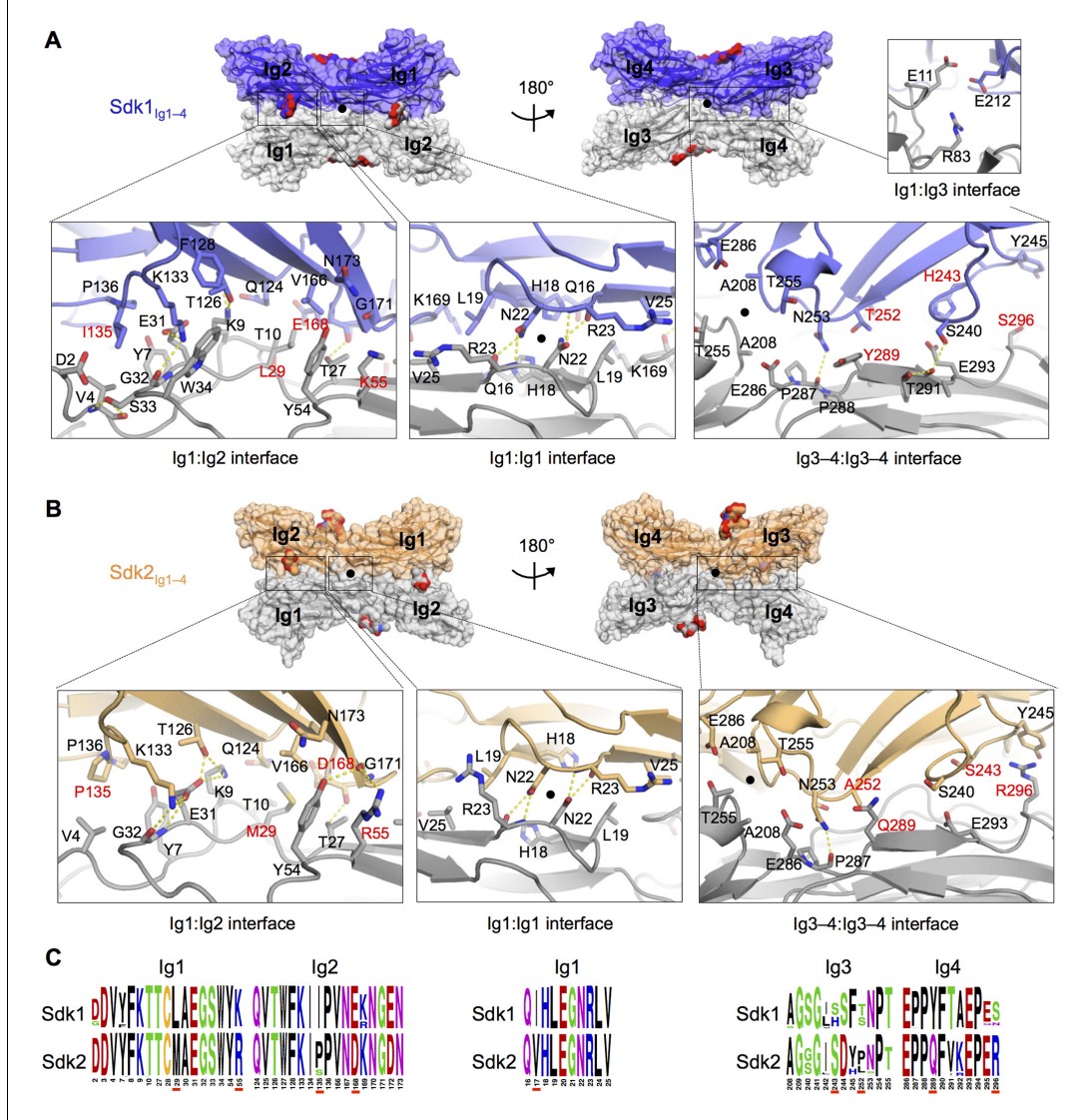

**Figure 4.** The Sdk1 and Sdk2 dimer interfaces are highly similar. (**A**) Sdk1 dimer interface. Top panel: Sdk1$_{Ig1-4}$ crystal form 2 dimer shown in surface view, one protomer dark blue, the other gray. Bottom panel: Close ups of the Ig1:Ig2, Ig1:Ig1, Ig3–4:Ig3–4 interfaces that make up the dimer. All interfacial residue side-chains, and main-chain atoms involved in hydrogen bonds, are shown in stick view. Residues labeled in red differ between Sdk1 and Sdk2. Hydrogen bonds are shown as yellow-dashed lines. The small Ig1:Ig3 interface observed only in the Sdk1$_{Ig1-4}$ crystal form 2 is shown in the top right. (**B**) Sdk2 dimer interface. Sdk2$_{Ig1-4}$ crystal form 1 dimer shown as for Sdk1 in **A**. One protomer is colored orange, the other gray. (**C**) Sequence logos of Sdk1 and Sdk2 dimer interface residues from the Ig1:Ig2, Ig1:Ig1, and Ig3–4:Ig3–4 interfacial regions generated from multiple sequence alignments of vertebrate Sdk1 and Sdk2 protein sequences (species used are listed in *Figure 4—source data 1*). Key interfacial residues showing differences between Sdk1 and Sdk2 are underlined in red.

The following source data and figure supplements are available for figure 4:

**Source data 1.** Protein amino acid sequences used to generate Sdk sequence logos.

**Figure supplement 1.** Sdk sequence logos showing conservation of the adhesive dimer interface residues and the Ig4:Ig5 intramolecular interface residues.

**Figure supplement 2.** Electrostatic surfaces of the Sdk dimerization interfaces.

**Figure supplement 3.** The crystal structure of Sdk2$_{Ig1-4}$ H18R/N22S mutant shows complete loss of the wild-type dimer interaction.

of the horseshoe dimer. This interaction is not seen in either of the Sdk2 structures, although the residues are conserved.

## Biophysical analysis of Sdk mutants reveals the relative contributions of the Ig1–2 and Ig3–4 regions to dimerization

We performed mutagenesis experiments to assess the contribution of the various contact regions in the crystallographically defined dimer. Mutating residues in the central Ig1:Ig1 interface (N22R in Sdk1 and N22R, N22S and H18R/N22S in Sdk2) resulted in the loss of dimerization for both Sdk1 and Sdk2 in AUC experiments (*Table 1*). These results—particularly the Sdk2 N22S mutation—highlight the critical importance of N22 and the two hydrogen bonds it forms to the dimer interaction. Consistent with this, a crystal structure of $Sdk2_{Ig1–4}$ H18R/N22S showed a loss of the back-to-back dimer interaction (*Figure 4—figure supplement 3*). Additionally, mutation of a salt-bridge residue (K133E) from the Ig1:Ig2 interface in Sdk1 resulted in 20-fold weaker binding in solution (*Table 1*).

Mutagenesis of a key Ig3–4:Ig3–4 interface residue (N253E) did not prevent Sdk dimerization, although it did reduce the dimer affinity, particularly for Sdk2, which went from 2.2 µM to 18.9 µM, implying a small Ig3–4 contribution to dimer strength of around 1.3 kcal/mol. Interestingly the $Sdk2_{Ig1–2}$/$Sdk1_{Ig3–4}$ chimera had a dimerization affinity intermediate to that of Sdk1 and Sdk2 (*Table 1*), suggesting that the Sdk2 Ig1–2 contributions to dimer affinity may be more than Sdk1 Ig1–2 while the Sdk1 Ig3–4 contributions to dimer affinity may be less than Sdk2 Ig3–4, and could therefore both underlie the lower dimer affinity of Sdk1 relative to Sdk2. The crystal structure of this chimera did not show an interaction between Ig3–4 regions in the dimer, although given the variability observed in the conformations of the dimers in the different Sdk1 and Sdk2 crystal structures, the Ig3–4 regions may interact in solution (*Figures 3B and 3C*).

## The Ig1–4 dimer is required for Sdk1-mediated cell adhesion

To determine whether the crystallographic dimer is the adhesive Sdk dimer, we tested the N22R mutation in the context of full length Sdk1 to see whether it would prevent Sdk-mediated cell adhesion in a cell aggregation assay. Aggregation of cells expressing the Sdk1 N22R mutant was significantly reduced compared to those expressing wild-type Sdk1 (*Figures 5A–B* for HEK cells and *Figure 5—figure supplement 1* for L cells). In addition Sdk1 N22K and L29E mutants also significantly reduced cell adhesion and the Sdk1 K133E mutant showed decreased cell aggregation in the same assay (*Figure 5B*). We also analyzed distribution of Sdk1 at sites of cell-cell contact. Wild-type Sdk1 localized to the cell-cell junctions, whereas the Sdk1 N22R mutant showed diffuse localization over the cell surface (*Figure 5C*) and showed no increased localization at sites of contact with cells that expressed wild-type Sdk1 (*Figure 5D*). These data indicate that the dimer observed both in solution and in the crystal structures is necessary for Sdk-mediated cell adhesion. The simplest conclusion from these data is that this dimer represents the adhesive dimer formed between molecules emanating from opposing cells (*trans* interaction).

## Sdk1 forms cis clusters dependent on the Ig1-4 dimer

The clustering of Sdk1 at sites of cell-cell contact is consistent with its role in intercellular adhesion. However, we found that Sdk1 also formed large punctate *cis* clusters (>0.25 µm²) on the surface non-interacting isolated cells (*Figure 5E* and *Figure 5—figure supplement 2A–D*). This clustering was not seen in cells expressing the Sdk1 N22R mutant; in these cells, Sdk1 N22R was more diffusely localized, forming only small clusters on the central region (*Figure 5E* and *Figure 5—figure supplement 2A–C*; both wild-type and mutant Sdk occasionally aggregated at the edges of cells). This difference did not result from different expression levels of wild-type and mutant Sdk1 or from different levels of protein reaching the cell surface (*Figure 5—figure supplement 2A,E*).These observations demonstrate that Sdk1 clusters are not dependent upon *trans* (cell-cell) interaction, but are dependent upon the identified horseshoe-mediated dimer interaction. This suggests that the horseshoe-mediated dimer can form *in cis*.

To assess the specificity of the *cis*-interaction, we generated constructs in which Sdk1, Sdk1 N22R or Sdk2 were fused to a fluorescent protein (RFP or YFP). We then clustered the Sdk1-RFP using a co-transfected nanobody (see Materials and Methods). When co-expressed, Sdk1-YFP co-clustered with the Sdk1-RFP, but Sdk2-YFP did not, indicating that Sdk-mediated *cis*-interactions, like Sdk-

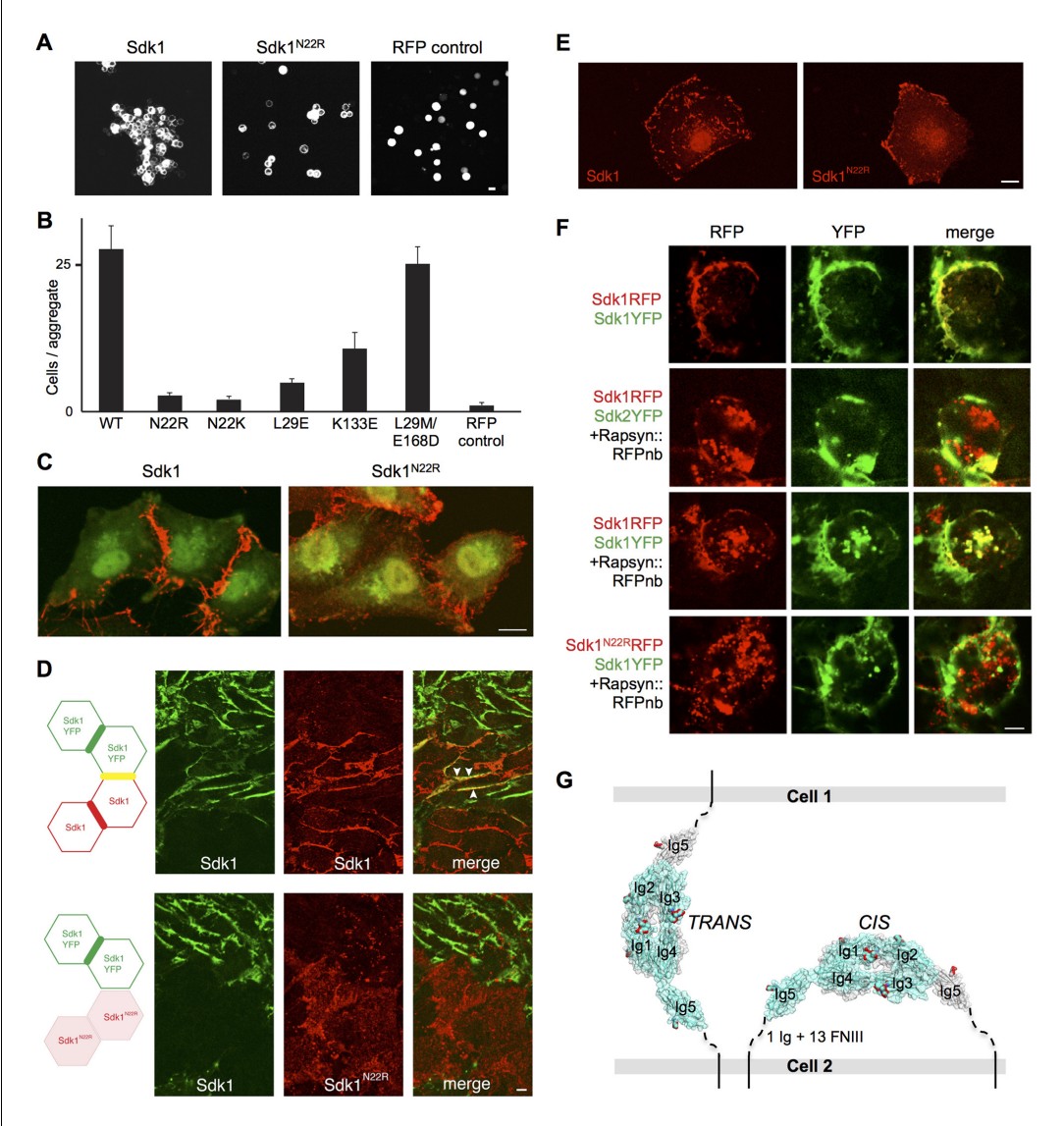

**Figure 5.** Sdk1 dimerization is required for cell aggregation and *cis* clustering. (**A**) Aggregation assay using N-cadherin deficient HEK-293 cells transiently transfected with wild-type (left panel) and N22R mutant (middle panel) Sdk1-RFP. Cytoplasmic RFP transfection was used as a negative control (right panel). (**B**) Quantification of the aggregation assay shown in **A** for wild-type (WT) Sdk1 and several Sdk1 dimer interface mutants (n = 15, mean ± S.E.). (**C**) Immunolabeling of Sdk1 (red) with a monoclonal antibody to Sdk1 in interacting L cells shows wild-type Sdk1 (left panel) localizes to the cell-cell junctions whereas the Sdk1 N22R mutant is diffusely localized (right panel). Counterstaining with wheat germ agglutinin (WGA, green) was used to visualize the cell surface. (**D**) Immunolabeled Sdk1 (red, stained with anti-Sdk1 cytoplasmic domain) and Sdk1YFP (green, Sdk1's cytoplasmic domain was replaced with YFP) co-localize at cell-cell junctions between Sdk1 and Sdk1YFP expressing L cells (top panels). Arrows indicate co-localization of red and green fluorescence. By contrast, Sdk1 N22R does not localize to cell-cell junctions between Sdk1 N22R and Sdk1YFP expressing cells (bottom panels). (**E**) Immunolabeling of Sdk1 with a monoclonal antibody to Sdk1 (red) in solitary L cells shows that wild-type Sdk1 localizes in puncta on the cell surface (left) whereas the Sdk1 N22R mutant is diffusely localized (right). (**F**) HEK-293 cells transiently transfected with both Sdk1RFP and Sdk1YFP express both proteins, which co-localize to cell-cell junctions (top row). Co-transfection of a Rapsyn::RFPnanobody induces clustering of Sdk1RFP away from cell-cell junctions (second and third rows). Sdk2YFP does not co-cluster with Sdk1RFP (second row), but Sdk1YFP does co-cluster (third row). However Sdk1YFP does not co-cluster with Sdk1 N22R-RFP/Rapsyn::RFPnanobody clusters (bottom row). (**G**) Our data suggest Sdk dimerizes using the crystallographically-determined interface, both between molecules emanating from opposing cell surfaces (in *trans*)—mediating cell-cell interactions—and between molecules emanating from the same cell surface (in *cis*)—mediating Sdk clustering. These interactions are shown schematically, using the Sdk1$_{Ig1-5}$ crystal structure to illustrate the dimer interaction. The remaining 1 Ig and 13 FNIII domains that constitute the rest of the Sdk extracellular domain are abbreviated to a dashed line, with the transmembrane and intracellular domains shown as solid lines. Scale bars in **A**, **C**, **D**, **E** and **F**, 5 µm.

*Figure 5 continued on next page*

*Figure 5 continued*

The following figure supplements are available for figure 5:

**Figure supplement 1.** Cell aggregation assay showing the N22R dimer interface mutant impairs Sdk1-mediated cell adhesion.

**Figure supplement 2.** Analysis of Sdk1 puncta.

mediated intercellular interactions, show homophilic specificity (*Yamagata and Sanes, 2008*; *Hayashi et al., 2005*; *Figure 5F*). Moreover, while wild-type Sdk1-YFP co-clustered with Sdk1-RFP, it did not co-cluster with Sdk1 N22R-RFP confirming that the dimer interaction is required for clustering (*Figure 5F* and *Figure 5—figure supplement 2F*). Altogether, our data are consistent with a model whereby the crystallographically-observed Ig1–4 dimer mediates homophilic interactions in *trans* (between cells), as well as in *cis* (between Sdks on the same cell surface) (*Figure 5G*).

## Sidekick specificity is primarily conferred by Ig1–2

Homophilic interactions of Sdk1 and Sdk2 promote specific synaptic connectivity in the retina (*Krishnaswamy et al., 2015*; M.Y., J.R.S., and A. Krishnaswamy, unpublished data). To assess the nature of this specificity in greater detail, we conducted in vitro SPR experiments in which we immobilized wild-type $Sdk1_{Ig1-6}$ and $Sdk2_{Ig1-6}$ onto an SPR chip by amine coupling, and then flowed a series of wild-type and mutant Sdk proteins over these surfaces to assess their relative binding. First, we flowed wild-type $Sdk1_{Ig1-6}$ and $Sdk2_{Ig1-6}$ over the $Sdk1_{Ig1-6}$ and $Sdk2_{Ig1-6}$ surfaces. Both constructs bound to both surfaces, indicating Sdk1 and Sdk2 can bind both homophilically and heterophilically. However, homophilic binding was stronger than the heterophilic binding for both the $Sdk1_{Ig1-6}$ and $Sdk2_{Ig1-6}$ surfaces (*Figure 6A*). This reveals that the segregation of Sdk1 and Sdk2 expressing cells into separate aggregates is determined not by a complete inability to interact heterophilically but rather a preference for homophilic interaction.

The results for $Sdk1_{Ig1-4}$ and $Sdk2_{Ig1-4}$ over the Ig1–6 SPR surfaces were comparable to those of the Ig1–6 proteins, indicating that it is the horseshoe portion of both Sdks that is responsible for both homophilic and heterophilic interaction (*Figure 6A*). The Ig1–2 interface mutants (Sdk1 N22R and Sdk2 H18R/N22S) showed a loss of both homophilic and heterophilic binding revealing that the heterophilic interaction observed here is mediated by the same interface as the homophilic interaction (*Figure 6A*). Consistent with the AUC results (*Table 1*), the Ig3–4 interface mutants (Sdk1

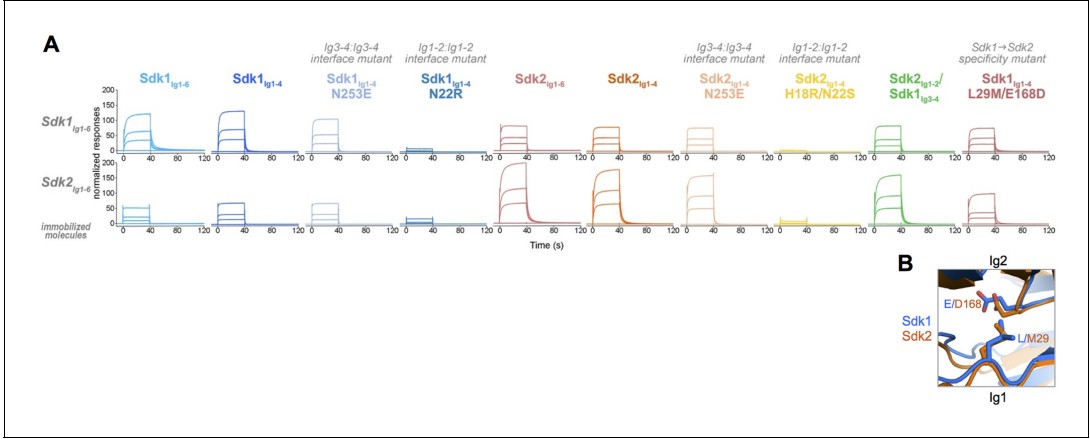

**Figure 6.** Sdk1 and Sdk2s' homophilic interactions are stronger than their heterophilic interaction. (**A**) Heterophilic and homophilic Sdk binding were analyzed by a surface plasmon resonance (SPR) experiment. $Sdk1_{Ig1-6}$ (top row) and $Sdk2_{Ig1-6}$ (bottom row) were covalently attached to the SPR chip and three different identical concentrations of each Sdk analyte (columns) were flowed over the $Sdk1_{Ig1-6}$ and $Sdk2_{Ig1-6}$ surfaces. The binding association and dissociation are shown by the normalized SPR response. (**B**) Close up of interacting specificity residues 29 and 168 in the Sdk1 (blue) and Sdk2 (orange) dimer interfaces.

N253E and Sdk2 N253E) showed slightly reduced binding relative to the wild-type homophilic binding. However, these mutations had no impact on heterophilic binding (*Figure 6A*).

To further define the basis of Sdk specificity, we generated a series of chimeric constructs converting Sdk1 to Sdk2 one Ig domain at a time in the Ig1–6 context for biophysical analysis and the full length context for cell aggregation assays. However, we were unable to produce these Ig1–6 chimeras at high enough solubility for accurate biophysical analysis, suggesting that the chimeras may not have fully native conformations. As noted above, a Sdk2$_{Ig1-2}$/Sdk1$_{Ig3-4}$ chimera was well behaved and formed a strong dimer in solution with comparable affinity to Sdk1$_{Ig1-4}$ and Sdk2$_{Ig1-4}$ (*Figure 3* and *Table 1*). Sdk2$_{Ig1-2}$/Sdk1$_{Ig3-4}$ behaved more like Sdk2 than Sdk1 in the SPR assay: It bound to the Sdk1$_{Ig1-6}$ surface similarly to Sdk2$_{Ig1-4}$, and more tightly to the Sdk2$_{Ig1-6}$ surface than Sdk1$_{Ig1-4}$ although not to the same level as Sdk2$_{Ig1-4}$ (*Figure 6A*). These results suggest that the majority of the Sdk1/Sdk2 specificity is localized to Ig1–2, but that Ig3–4 may also contribute to specificity. However, given the poor behavior of the other Sdk chimeras we made, we cannot rule out the possibility that the small differences in behavior between the Sdk2$_{Ig1-2}$/Sdk1$_{Ig3-4}$ chimera and Sdk2$_{Ig1-4}$ are due to slight imperfections in the chimeric molecule.

Analysis of the crystal structures revealed that there are two Ig1–2:Ig1–2 interfacial residues that interact with one another and show conserved differences between Sdk1 and Sdk2 (*Figure 4* and *Figure 4—figure supplement 1*). These residues—29$_{Ig1}$ and 168$_{Ig2}$—therefore seem good candidates for being involved in specificity. In Sdk1, L29 interacts with the aliphatic portion of the E168 side chain, with the charged carboxylate group extending away from the hydrophobic leucine. In Sdk2 the M29 side chain interacts with D168, again with the carboxylate extending away (*Figure 6B*). Although these are not normally favorable residue pairs, in both cases the interactions are primarily between carbon atoms so there may be some hydrophobic contribution to affinity from this interaction. Mutation of these residues showed that the Sdk2 M29/D168 interaction is more favorable (or possibly less unfavorable) than the Sdk1 L29/E168 interaction, since the Sdk1$_{Ig1-4}$ L29M/E168D double mutant has a greater dimerization affinity than wild-type Sdk1$_{Ig1-4}$ (*Table 1*). Similar to wild-type, the double mutant also mediates cell-cell adhesion (*Figure 5B*).

Not only did the Sdk1$_{Ig1-4}$ L29M/E168D mutant shift the homodimerization affinity of Sdk1 towards that of Sdk2 (*Table 1*), it also modified the specificity of Sdk1 to more closely mimic that of Sdk2. Specifically, this mutant showed weaker binding to the Sdk1$_{Ig1-6}$ surface relative to wild-type Sdk1$_{Ig1-4}$ and stronger binding to the Sdk2$_{Ig1-6}$ surface (*Figure 6A*). However the Sdk1$_{Ig1-4}$ L29M/E168D mutant did not bind to the Sdk2$_{Ig1-6}$ surface as strongly as the Sdk2$_{Ig1-2}$/Sdk1$_{Ig3-4}$ chimera or wild type Sdk2 $_{Ig1-4}$, indicating that additional Ig1–2 residues also contribute to the observed specificity, alongside the putative Ig3–4 contribution. As already noted above, there are three additional Ig1–2 interfacial residues that show conserved differences between Sdk1 and Sdk2 (I/V17, K/R55, I/P135), which may contribute to specificity. Moreover, non-interfacial residues and interdomain flexibility have also been shown to play a role in determining specificity between related molecules (*Vendome et al., 2014*). Although we have shown here that residues at positions 29 and 168 contribute to the differences between Sdk1 and Sdk2, there are clearly many other factors that can contribute. Teasing out individual contributions as we did for N- and E-cadherin (*Vendome et al., 2014*) will require additional work.

## Discussion

The cell adhesion molecules Sdk1 and Sdk2, like Dscams and CNTNs, provide molecular cues that determine the specificity of particular synaptic connections in the retina (*Yamagata and Sanes, 2008*). Here, we have presented multiple crystal structures of the Sdk1 and Sdk2 Ig-domain mediated homodimers that are required for Sdk-mediated cell-cell adhesion. The dimers are formed by interaction between the Ig1–4 horseshoe regions of the Sdks. The dimers are primarily mediated by anti-parallel interactions between the Ig1–2 portions of the horseshoes, and mutagenesis studies demonstrated that this interaction was necessary for Sdk dimerization in solution and Sdk-mediated cell-cell adhesion. The Ig3–4 halves of the horseshoes, which interact in an anti-parallel manner in some but not all of the Sdk1 and Sdk2 crystal structures, only contribute a small amount to the binding energy. A previous study proposed Ig2 beta strand residues Q147–A152 were involved in the Sdk adhesive interaction (*Hayashi et al., 2005*), however our structures show that this strand is not located in the dimer interface. *Hayashi et al. (2005)* sought to demonstrate the involvement of this

beta strand by deletion of these residues, which would dramatically affect the fold of Ig2, explaining the loss of cell-cell adhesion they observed in cell expressing this mutant.

The horseshoe-motif mediated Sdk dimers presented here are very different from the horseshoe-motif mediated homodimers of Dscam1, neurofascin, and CNTN, for which there are published crystal structures (*Figure 3—figure supplement 1*). The human neurofascin and human CNTN2 horseshoe-motif dimers are both primarily mediated by anti-parallel interactions between the Ig2 G strands, with limited involvement from Ig1 and none from Ig3 and Ig4 (*Figure 3—figure supplement 1A*; *Liu et al., 2011*; *Mörtl et al., 2007*). The *Drosophila* Dscam1 dimer is mediated by horseshoe-motif domains Ig2 and Ig3, alongside Ig7 (*Figure 3—figure supplement 1C*; *Wojtowicz et al., 2004*; *Sawaya et al., 2008*). The horseshoe portion of this interaction involves Ig2:Ig2 and Ig3:Ig3 interactions, and is not sufficient for dimerization alone (*Meijers et al., 2007*), with the Ig7:Ig7 interaction also required (*Wojtowicz et al., 2007*; *Sawaya et al., 2008*). Therefore, different IgSF members utilize different horseshoe-motif surfaces to mediate homophilic interactions, highlighting the versatility of the motif. Heterophilic interactions have also been reported for horseshoe-motif proteins, including binding of CNTNs 1, 3, 4, 5 and 6 to receptor protein tyrosine phosphatase gamma (PTPγ) (*Bouyain and Watkins, 2010*). The crystal structure of CNTN4 with the carbonic anhydrase-like domain from PTPγ shows that horseshoe domains Ig2 and Ig3 mediate this interaction, utilizing the same face for interaction as the Dscam1 homodimer (*Bouyain and Watkins, 2010*). To date no heterophilic interactions have been reported for the Ig domain regions of Sdk, however the opening of the Ig3–4 regions in the Sdk dimers could in principle accommodate interaction with another molecule, or potentially the FNIII portion of the Sdk extracellular region. Indeed, the flexibility of the Sdk dimer interaction that we have observed in our Sdk1 and Sdk2 crystal structures is surprising and could be explained by the need to accommodate an additional interaction.

As discussed above, studies in cells reveal that Sdk1 forms large *cis* clusters independent of contacts with other cells. Disrupting Sdk1 Ig1–4-mediated dimerization by mutation of a key interfacial residue (N22R) resulted in a loss of this Sdk clustering in isolated cells. Instead the Sdk1 N22R mutant showed a more diffuse expression across the whole cell surface, with only small clusters evident (*Figure 5E* and *Figure 5—figure supplement 2*). This indicates that the Sdk1 Ig1–4-mediated dimer is required for Sdk clustering in large aggregates. For some cell surface molecules such as classical cadherins, binding *in cis* through the cell-cell recognition interface is geometrically unfavorable due to the rigidity of the interacting molecules. However, since in Sdks there are 13 FNIII domains and two additional Ig domains separating the horseshoe 'dimerization' region from the membrane, the molecules are sufficiently flexible such that there should be no geometric constraints preventing the formation of a *cis* dimer. The observed Ig1–4-dimer-dependent *cis*-clusters are clearly larger assemblies than dimers. Moreover, Sdks still cluster after replacing their native cytoplasmic domain with YFP or RFP, demonstrating that intracellular sequences are dispensable. Therefore, additional interactions involving the FNIII, or Ig domain regions are also likely to contribute to their formation. In addition the presence of small aggregates in the N22R mutant cells could indicate that such additional *cis* interactions may form independent of the horseshoe-mediated dimer. However, the nature of these interactions has not yet been identified.

When we mixed Sdk1 and Sdk2 expressing cells in our cell assays, we observed few mixed aggregates. In contrast, in solution Sdk Ig1–4 proteins dimerize both homophilically and heterophilically although the homophilic interactions are significantly stronger. We suggest three possible explanations for this apparent discrepancy. First, cell aggregation depends on the affinities of adhesion molecules; so strong homophilic interactions will lead to the formation of separate aggregates even when weaker heterophilic interactions can also occur (*Katsamba et al., 2009*). Second, we cannot exclude the possibility that determinants proximal to Ig1–4 enhance homophilic or attenuate heterophilic interactions. Third, and most interesting, competition between *cis* and *trans* interactions could sharpen specificity. The Sdk Ig1–4-mediated dimer is required for both Sdk *cis*-clustering in isolated cells and Sdk-mediated cell-cell adhesion. Since both interactions are mediated by the same interface, *cis* and *trans* dimers are likely to be mutually exclusive. Thus, *cis*-dimers formed between Sdks on the same cell surface would need to dissociate in order for Sdk dimers to form *in trans* between molecules on adjacent cells (*Figure 7*). Since homophilic interactions are stronger than heterophilic interactions, heterophilic *trans* interactions would not be able to outcompete homophilic *cis* interactions, ensuring that no adhesion will be observed between cells containing different Sdks. An

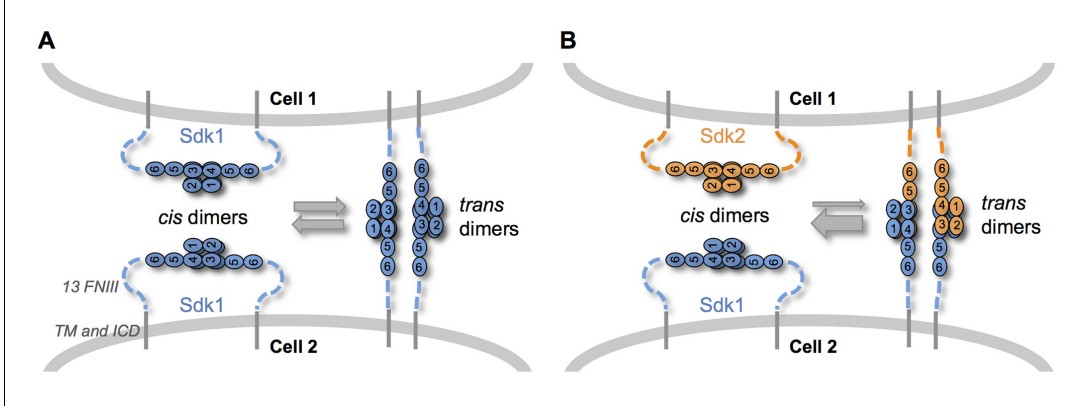

**Figure 7.** Sdk specificity could be accentuated by competition between *cis* and *trans* interactions. (**A**) Schematic of two neighboring cells expressing Sdk1 showing the competition between homophilic Ig1–4-mediated dimer interactions *in cis* and *in trans*. The six immunoglobulin domains (Ig) are represented as numbered ovals and the thirteen extracellular fibronectin type III (FNIII), whose arrangement is unknown, are represented as dashed lines. (**B**) Schematic of two neighboring cells, one expressing Sdk1 and one expressing Sdk2, showing the competition between the stronger homophilic dimer interactions *in cis* and the weaker heterophilic dimer interactions *in trans*.

analogous logic is used to enhance binding specificity in classical cadherins which dimerize through a strand swapping mechanism (*Chen et al., 2005*; *Vendome et al., 2011*; *Vendome et al., 2014*)

*Cis*-clustering and *cis/trans* competition could enhance specific connectivity in the crowded, complex neuropil of the central nervous system. For example, although most Sdk-positive retinal neurons express either Sdk1 or Sdk2, one population expresses both Sdks (*Krishnaswamy et al., 2015*). *Cis*-clustering could lead to the formation of distinct Sdk1 and Sdk2 subdomains, which could engage different synaptic partners. Moreover, at least in the retina, cells that express only one Sdk arborize in close proximity to cells that express the other Sdk (*Krishnaswamy et al., 2015*). *Cis/trans* competition could play an important role in preventing the formation of inappropriate contacts between these populations.

In summary, the homophilic specificities of Sdk1 and Sdk2 play a crucial role in defining the patterning of synaptic connections in the retina. Our data suggest that these cell-patterning effects are based on the relative affinity of homophilic and heterophilic binding in the context of competition between *cis* and *trans* Sdk interactions. Further work will be necessary to demonstrate this in a neuronal context, as well as to determine the functional role of Sdk *cis* clustering.

## Materials and methods

### Protein expression and purification

cDNA sequences for the Ig1–4 (residues 1–379), Ig1–5 (residues 1–476) and Ig1–6 (residues 1–569) portions of the mouse Sdk1 (NCBI NM_177879) and Sdk2 (NCBI NM_172800) extracellular regions (excluding the native signal sequence) were cloned into the mammalian expression vector pαSHP-H mammalian expression vector (a kind gift from Daniel J. Leahy, John Hopkins University), which contains the DNA sequences for the PTPa signal sequence, N-terminal hexahistidine and Strep tags, and the cleavage recognition site for PreScission protease. Point mutations were generated by the Quikchange method (Stratagene, CA) using the KOD Hot Start DNA polymerase (Novagen, END Millipore, MA). The Sdk2$_{Ig1–2}$/Sdk1$_{Ig3–4}$ chimera (Sdk2 residues 1–188 and Sdk1 residues 189–379) was assembled by PCR.

Constructs were transfected into the adhesive human embryonic kidney (HEK)-293 GNTi- cells (Sdk1$_{Ig1–4}$ for crystallization) or suspension HEK-293F cells (all other proteins) (Invitrogen, Carlsbad, CA) with phenylethyleneimine (Polysciences Inc., Warrington, PA). Conditioned medium was collected after 5–6 days. The secreted Sdk proteins were purified by nickel-affinity chromatography—followed by anion and cation exchange chromatography for Sdk1$_{Ig1–4}$ from GNTI- cells—and, after cleavage of the tag using PreScission protease (Invitrogen, Carlsbad, CA), size exclusion

chromatography using an Akta FPLC system (GE Healthcare, Pittsburgh, PA). The purified proteins were concentrated to 3–8.5mg/ml in 10 mM Tris-Cl pH8, 150 mM sodium chloride (all Sdk1 proteins) or 10 mM Bis-Tris pH6, 150 mM sodium chloride (all Sdk2 proteins and the Sdk2$_{Ig1-2}$/Sdk1$_{Ig3-4}$ chimera). Purified PreScission-cleaved proteins N-terminal residues are GPALA for Sdk1 and GPAGA for Sdk2, followed by the predicted mature N-termini: QDD for both Sdk1 and Sdk2. Residue numbering is from this glutamine (Q1). The mature N-termini were predicted using the SignalP 4.0 server (*Petersen et al., 2011*).

## X-ray crystallography

Crystals were grown in 1–2μL drops using the vapor-diffusion method at 22°C. Crystallization conditions were as follows, with added cryoprotectants given in parentheses: Sdk1$_{Ig1-4}$ crystal form 1, 12–16% (w/v) PEG4000, 0.2 M ammonium citrate (30% (v/v) ethylene glycol, with 0.2 M cesium iodide); Sdk1$_{Ig1-4}$ crystal form 2, 10% PEG8000, 0.01 M zinc chloride, 0.1 M HEPES, pH 7.5 (30% (v/v) ethylene glycol); Sdk1$_{Ig1-5}$, 1.4 M ammonium sulfate, 0.5 M lithium chloride, 10 mM yttrium(III) chloride (30% (v/v) glycerol); Sdk2$_{Ig1-4}$ crystal form 1, 1.5 M ammonium sulfate, 0.1 M sodium citrate pH 4 (30% (v/v) glycerol); Sdk2$_{Ig1-4}$ crystal form 2, 24% PEG3350 (w/v), 0.2 M sodium chloride, 0.1 M Bis-Tris pH 5.5 (20% (v/v) ethylene glycol); Sdk2$_{Ig1-4}$ H18R/N22S, 13.5% PEG3350 (w/v), 0.1 M ammonium sulfate, 0.2 M sodium dihydrogen phosphate (25% (v/v) glycerol); Sdk2$_{Ig1-2}$/Sdk1$_{Ig3-4}$, 3% (w/v) PEG8000, 40% (v/v) 2-methyl-2,4-pentanediol, 0.1 M sodium cacodylate pH 6.9.

Diffraction data were collected from single crystal flash frozen at 100 K on the X4A and X4C beamlines at the National Synchrotron Light Source, Brookhaven National Laboratory or the Northeastern Collaborative Access Team beamlines 24-ID-C/E at the Advanced Photon Source, Argonne National Laboratory. The data for both Sdk1$_{Ig1-4}$ crystal forms were processed using HKL2000 (*Otwinowski and Minor, 1997*) and scaled/merged with SCALEPACK (*Otwinowski and Minor, 1997*); all other data were processed with iMOSFLM (*Battye et al., 2011*) and scaled/merged with SCALA or AIMLESS (*Evans, 2006*; *Evans and Murshudov, 2013*). Sdk1$_{Ig1-4}$ crystal form 1 was solved by single wavelength anomalous diffraction (SAD) phasing method using crystals cryo-soaked with iodide ions. Substructure solution, phasing, and density modification were carried out using AUTOSOL in PHENIX (*Adams et al., 2010*); model building and refinement were carried out using COOT (*Emsley et al., 2010*) and PHENIX respectively. The other structures were solved by molecular replacement with PHASER (*McCoy et al., 2007*) using the Sdk1$_{Ig1-4}$ structure as a search model. Iterative model building and refinement were carried out using COOT and PHENIX, to yield the final refined structures whose statistics are detailed in *Figure 2—source data 1*.

## Structural analysis

Protein interface buried surface areas were obtained using the 'protein interfaces, surfaces and assemblies' service (PISA) from the European Bioinformatics Institute (http://www.ebi.ac.uk/pdbe/prot_int/pistart.html; *Krissinel and Henrick, 2007*). Interdomain angles were calculated using UCSF Chimera (*Pettersen et al., 2004*). Root mean square deviations over aligned Cα atoms between structures were calculated using Pymol (Schrödinger, LLC, New York, NY). Crystal structure figures were made using Pymol or UCSF Chimera.

## Electrostatic potential calculations

Prior to the electrostatic potential calculation, the Sdk1$_{Ig1-5}$ structure and Sdk2$_{Ig1-5}$ structure/model were prepared as follows. The Sdk2$_{Ig1-4}$ crystal form 1 structure was used with an Sdk2 Ig5 model, which was generated with MODELLER (*Webb and Sali, 2014*) using Sdk1$_{Ig1-5}$ as a template. Missing segments from both structures were also built using the MODELLER program, using the other chain or another crystal form structure as a template. Hydrogen atoms and missing side chain atoms were built with the CHARMM program (*MacKerell et al., 1998*). The structures were then subjected to a two-step minimization (conjugate gradient method) implemented in NAMD (*Phillips et al., 2005*) with the CHARMM force field (*MacKerell et al., 1998*). In the first minimization step, hydrogen atoms were minimized for 3000 steps with strong harmonic constraints of 50 kcal/mol Å$^2$ applied to non-hydrogen atoms. In the second step, strong harmonic constraints were applied to the backbone atoms and to all Asn173 atoms, which is covalently attached to NAG, while side chain atoms were

minimized with constraints for 5000 steps. The interfacial NAG was subsequently added to the minimized structures.

Electrostatic potentials were obtained by solving the Poisson–Boltzmann equation using finite difference methods as implemented in the DelPhi program (*Honig and Nicholls, 1995*). Atomic radii and charges were taken from CHARMM (*MacKerell et al., 1998*). The interior of the proteins and water were modeled as dielectric media with dielectric constants of 2 and 80, respectively. Ionic strength was set to 0.145 M and an ion exclusion radius of 2 Å was used. The numerical calculation of the potential was iterated to convergence, defined as the point at which the potential changes $<10^{-5}$ kT e$^{-1}$ between successive iterations. DelPhi calculations were run on a cubic lattice with four focusing steps of increasing resolution (from 0.5 to 2.6 grids per Å). Visualization of electrostatic surfaces was carried out with UCSF Chimera (*Pettersen et al., 2004*).

### Sequence conservation logos

Orthologs for Sdk1 and Sdk2 were identified using NCBI BLAST (*Altschul et al., 1997*). The species for which orthologs were identified are listed in *Figure 4—source data 1*. Multiple sequence alignments of the Sdk1 and Sdk2 orthologs were generated using COBALT (*Papadopoulos et al., 2007*) and sequence logos were generated using WebLogo3 (*Crooks et al., 2004*).

### Sedimentation equilibrium analytical ultracentrifugation measurements

Experiments were performed in a Beckman XL-A/I analytical ultracentrifuge (Beckman-Coulter, Palo Alto CA, USA), utilizing six-cell centerpieces with straight walls, 12 mm path length and sapphire windows. All proteins were dialyzed overnight and then diluted to appropriate concentration in dialysis buffer. For all Sdk1 proteins 150 mM sodium chloride, 10 mM Tris-Cl pH 8.0 were used, and for all Sdk2 proteins and the Sdk2$_{Ig1-2}$/Sdk1$_{Ig3-4}$ chimera 150 mM sodium chloride, 10 mM Bis-Tris pH 6.0 were used. Samples were diluted to an absorbance at 10 mm and 280 nm of 0.65, 0.43 and 0.23 in channels A, B and C, respectively. Dilution buffer was used as blank. All samples were run in duplicate at four speeds. Ig1–5 and Ig1–6 constructs were run at 9000, 11,000, 13,000 and 15,000 rpm, and Ig1–4 constructs were run at 12,000, 16,000, 20,000 and 24,000 rpm except Sdk2$_{Ig1-4}$ N22S and N22R, which were run at 11,000, 14,000, 17,000, 20,000 rpm. The lowest speed was held for 20 hr after which four UV-scans were taken with 1 hr interval, the second lowest speed held for 10 hr, followed by four scans as above, the third lowest and the highest speed performed identically as the second lowest speed. All measurements were done at 25°C, and detection was by UV at 280 nm. Solvent density and protein v-bar were determined using the program SednTerp (Alliance Protein Laboratories, Corte Cancion, Thousand Oaks, CA, USA). Intact molecular weights of the purified Sdk proteins were obtained using matrix-assisted laser desorption/ionization (MALDI) mass spectrometry conducted by the Columbia University mass spectrometry facility. For calculation of dimeric $K_d$ and apparent molecular weight, all useful data were used in a global fit, using the program HeteroAnalysis, obtained from the University of Connecticut (www.biotech.uconn.edu/auf).

### Surface plasmon resonance (SPR) binding experiments

SPR binding assays were performed using a Biacore T100 biosensor equipped with a Series S CM4 sensor chip (GE Healthcare, Pittsburgh, PA). Mouse Sdk1$_{Ig1-6}$ and Sdk2$_{Ig1-6}$ were immobilized over independent flow cells using amine-coupling chemistry in HBS pH 7.4 (10 mM HEPES, 150 mM NaCl) buffer at 35°C using a flow rate of 20 μL/min. Dextran surfaces were activated for 10 min using equal volumes of 0.1 M NHS (*N*-Hydroxysuccinimide) and 0.4 M EDC (1-Ethyl-3-(3-dimethylaminopropyl) carbodiimide). Each protein was immobilized at 65 μg/mL in 10 mM sodium acetate, pH 4.5, for 30 s. The immobilized surface was blocked using a 4-minute injection of 1.0 M ethanolamine, pH 8.5. Approximately 2000 RU of each Sdk1$_{Ig1-6}$ and Sdk2$_{Ig1-6}$ was immobilized over each flow cell. An unmodified surface was used as a reference surface to correct for bulk refractive index shifts. Binding analysis was performed at 25°C in a running buffer of 10 mM Tris-HCl, pH 8.0, 150 mM NaCl, 0.25 mg/mL BSA and 0.005% (v/v) Tween-20. Each protein was prepared in buffer at 30, 10, and 3.3 μM (a three-fold dilution series). Each binding cycle consisted of a 40-second association phase and a 90-second dissociation phase at a flow rate of 50 μL/min, followed by a 60-second buffer wash at 100 μL/min. Every protein concentration was tested in duplicate within the same experiment. Buffer cycles were performed before and after each concentration series to double reference the sample

binding signals to correct for systematic noise and instrument drift. The data were processed using Scrubber 2.0 (BioLogic Software, Campbell, Australia). Binding responses were normalized for molecular weight differences between the proteins.

## Cell assay constructs and cell lines

Mouse Sdk1 and Sdk2 cDNA were cloned under pCMV promoters (Clontech, Mountain View, CA), and fused to a sequence of yellow fluorescence protein (YFP) or mCherry (RFP). We codon-optimized the mouse Sdk1 sequence since the N-terminus of mouse Sdk1 cDNA possesses a GC-rich region which was refractory to PCR. Chimeric constructs were generated by standard molecular cloning using restriction enzymes and Q5 DNA polymerase-assisted PCR (NEB, Ipswich, MA), or the Gibson assembly kit (SGI-DNA, La Jolla, CA). Mutants were generated using the QuikChange site-directed mutagenesis kits (Agilent Technologies, Santa Clara, CA). A construct for overexpressing a Rapsyn::RFPnanobody fusion protein (Rapsyn::RFPnb) was generated by synthesizing a construct encoding the 90 amino acid self-association domain of Rapsyn (*Ramarao and Cohen, 1998*) and a mCherry-binding nanobody LaM-4 (*Fridy et al., 2014*) in a pCMV backbone.

HEK-293T and L cells were obtained from the American Type Culture Collection (ATCC; Manassas, VA). Because HEK-293T cells are occasionally reported to be contaminated with HeLa cells (International Cell Line Authentication Committee; http://iclac.org/databases/cross-contaminations/) we confirmed that the cells were G418 resistant, a characteristic of HEK-293T but not HeLa cells. Contamination has not been reported for L cells. Cells were cultured in DMEM supplemented with 10% fetal calf serum and penicillin/streptomycin plus Normocin (Invivogen, San Diego, CA) for L cells and both Normocin and G418 (Invivogen, San Diego, CA) for HEK-293T cells. The morphological appearance of both cell types corresponded to previously published descriptions.

To generate L cell lines stably expressing mouse Sdk1, Sdk2, their mutants, and fluorescent protein derivatives, the sequences were cloned into a piggyBac transposon vector pXL-CAG-Zeocin-3xF2A (*Martell et al., 2016*). L cells were transfected with the appropriate Sdk construct together with a piggyBac transposase vector pCAG-mPBorf (*Yamagata and Sanes, 2012a*) using DMRIE-C (Invitrogen, Carlsbad, CA), trypsinized after 2 days, replated into larger plates, and selected with 1 mg/mL Zeocin (Invivogen, San Diego, CA) for 2–3 weeks. Surviving colonies were transferred to new plates and screened with antibodies against Sdk or fluorescence to select clones with high and homogeneous expression. In some experiments, Zeocin-selected stable cells were pooled, and stained with antibodies.

HEK-293T cells endogenously express N-cadherin (CDH2) which results in some background in cell adhesion assays. To decrease this background, expression of N-cadherin was fully eliminated by disrupting both alleles of the N-cadherin gene using CRISPR-mediated gene disruption. Full characterization of the N-cadherin-deficient HEK-293 cell line will be described elsewhere (M.Y. and J.R.S. unpublished). HEK-293T cells were transfected with DMRIE-C in OptiMEM (Invitrogen), and used for experiments 2–3 days after transfection. For co-transfection into 293T cells, two Sdk plasmids and a Rapsyn::RFPnb plasmid were mixed in a 5:5:1 ratio.

## Antibodies and immunostaining

Production of affinity-purified rabbit polyclonal antibodies to the cytoplasmic domain of mouse Sdk1 was described previously (*Krishnaswamy et al., 2015*). A mouse monoclonal antibody to mouse Sdk1 was generated from a Sdk1-knockout mouse (*Krishnaswamy et al., 2015*) immunized with mouse Sdk1-expressing L cells. Briefly, splenocytes from a hyperimmunized mouse were fused to a myeloma cell line FOXNY (ATCC), selected, and screened by immunostaining Sdk1-transfected L cells. One established hybridoma line, MS1-7, produces an IgG1/kappa monoclonal antibody that recognizes FNIII domains of mouse Sdk1 (not shown). Chicken anti-GFP antibodies were described previously (*Yamagata and Sanes, 2012b*). Species-specific Alexa dye-conjugated secondary antibodies were obtained from Jackson ImmunoResearch (West Grove, PA). Fluorescein-conjugated wheat germ agglutinin (WGA) was from Vector Laboratories (Burlingame, CA). Cultured cells on glass coverslips (Bellco Glass, Vineland, NJ) were fixed with 4% paraformaldehyde/PBS for 30 min at 4°C, and then either observed directly after mounting with Fluorogel (Electron Microscopy Sciences, Hatfield, PA), or immunostained after treatment with 0.1% (w/v) TritonX-100/PBS for 10 min at room temperature. The standard immunostaining procedure used was described previously

(*Yamagata and Sanes, 2012a*). Images were analyzed using a PlotProfile or the Analyze Particles plug-in of Image-J (version 1.47d, Fiji). For immunostaining of Sdk1 on the cell surface, cells were incubated with DMEM plus 10% (v/v) fetal calf serum supplemented with purified 1 µg/ml MS1-7 antibody (IgG1) for 30 min at 4°C, rinsed with the same medium, fixed with 4% paraformaldehyde/ PBS, stained with Cy3-conjugated secondary antibodies, rinsed, mounted, and observed under a confocal microscope. Alternatively, the bound antibodies were quantified using a colorimetric enzyme-linked immunosorbent assay. Briefly, after incubating live cells with MS1-7, the paraformal- dehyde-fixed cells were treated with 0.3% $H_2O_2$/PBS for 30 min at room temperature, blocked with 5% (w/v) skimmed milk (BioRad)/PBS for 30 min, incubated with peroxidase-conjugated goat anti- mouse immunoglobulins (Jackson ImmunoResearch, 1:1000 dilution in 0.5% BSA/PBS) for 2 hr, rinsed with PBS, and developed with o-phenylenediamine/$H_2O_2$.

### Cell aggregation assay

Confluent stably-transfected L cells or transiently-transfected 293T cells were trypsinized in the pres- ence of 1 mM EDTA at 37°C as described previously (*Yamagata and Sanes, 2008*). In some experi- ments, cells were labeled with green or red Cell Trackers (Invitrogen). The reaction was stopped by adding the same volume of 0.1 mg/ml soybean trypsin inhibitor (T6522, Sigma, St. Louis, MO) and 10 µg/ml deoxyribonuclease I (DN25, Sigma) in HBSS supplemented with 20 mM HEPES, pH 7.4. All the cell aggregation assays were carried out in 24-well non-tissue culture plasticwares that had been precoated with 0.5% BSA/HBSS. In each well, dissociated cells were mixed with 1 ml of HBSS con- taining 0.5% (w/v) BSA, 1 µg/ml deoxyribonuclease I, 20 mM HEPES, pH 7.4, and rotated at room temperature for 30–60 min. The reaction was stopped by adding 1 ml of 4% (w/v) paraformalde- hyde/PBS, and observed under fluorescent microscopes.

### Accession numbers

The atomic coordinates and structure factors for the reported crystal structures are deposited in the Protein Data Bank under accession codes PDB: 5K6U, 5K6V, 5K6W, 5K6X, 5K6Y, 5K6Z, and 5K70.

## Acknowledgements

We thank John Schwanof and Randy Abramowitz at Brookhaven National Lab XSLS X4A/C beam- lines and Igor Kourinov, Surajit Banerjee, and Narayanasami Sukumar at NE-CAT Advanced Photon Source 24-ID-C/E beamlines, Argonne National Laboratory (NIH grant P41 GM103403) for their sup- port with synchrotron data collection. We thank Rotem Rubinstein (Columbia) for the electrostatics calculations. We thank Matilda Lynton (Brown University) for help with crystallization.

## Additional information

### Funding

| Funder | Grant reference number | Author |
|--------|------------------------|--------|
| Howard Hughes Medical Insti- tute | | Xiangshu Jin Phinikoula S Katsamba Göran Ahlsén Alina P Sergeeva Barry Honig |
| National Science Foundation | MCB-1412472 | Barry Honig |
| National Institutes of Health | NS029169 | Joshua R Sanes |
| National Institutes of Health | EY022073 | Joshua R Sanes |
| National Institutes of Health | R01GM062270 | Lawrence Shapiro |

The funders had no role in study design, data collection and interpretation, or the decision to submit the work for publication.

## Author contributions
KMG, Designed research, Conducted protein production and crystallography experiments, Analyzed and interpreted data and wrote the manuscript; MY, Designed research, Conducted cell-based assays, Analyzed and interpreted data and wrote the manuscript, ; XJ, Designed research, Determined the Sdk1 Ig1–4 crystal structures and analyzed data; SM, Produced proteins for crystallography and biophysical experiments; PSK, Performed and analyzed the SPR experiments; GA, Performed and analyzed the AUC experiments; APS, Performed bioinformatics analysis of horseshoe-motif proteins; BH, LS, Designed research, Analyzed and interpreted data and wrote the manuscript; JRS, Designed research, Analyzed and interpreted data and wrote the manuscript, Contributed unpublished essential data or reagents

## Author ORCIDs
Masahito Yamagata, http://orcid.org/0000-0001-8193-2931
Barry Honig, http://orcid.org/0000-0002-2480-6696
Joshua R Sanes, http://orcid.org/0000-0001-8926-8836
Lawrence Shapiro, http://orcid.org/0000-0001-9943-8819

## Additional files

### Supplementary files
• Supplementary file 1. Sidekick and other horseshoe IgSF protein characteristics. (A) Sidekick inter-domain angles. (B) Structural comparison of IgSF protein horseshoe structures.

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
