## [Decision Letter]

Thank you for submitting your article "Molecular basis of sidekick-mediated cell-cell adhesion and specificity" for consideration by *eLife*. Your article has been favorably evaluated by John Kuriyan as the Senior Editor and three reviewers, one of whom, Mingjie Zhang (Reviewer #1), is a member of our Board of Reviewing Editors, and another is E Yvonne Jones (Reviewer #3).

The reviewers have discussed the reviews with one another and the Reviewing Editor has drafted this decision to help you prepare a revised submission. We hope you will be able to submit the revised version within two months, so please let us know if you have any questions first.

Summary:

Sidekick 1 and Sidekick2 (Sdk1and Sdk2) are Ig superfamily proteins which regulate the formation of neural circuits. Previous studies have shown that Sdk1 and Sdk2 exhibit preferential homophilic binding. That is, heterophilic binding occurs but homophilic binding is stronger. The role of these two paralogs has been characterized best in the context of layer-specific connections in the chick and mouse inner plexiform layer of the retina. Here Sdk1 and Sdk2 act with other Ig superfamily proteins, Dscam and DscamL1, and various contactins to regulate layer-specific connectivity. In this paper, Goodman et al. describe the molecular basis of Sdk-mediated cell adhesion. The paper includes several crystal structures, biophysical studies using analytical ultracentrifugation and Biocore to assess binding quantitatively and cell adhesion experiments to explore the interaction of the proteins in the context of living cells.

The ectodomains of Sdk1 and Sdk2 comprise, from the N-terminus, 6 Ig domains and 13 FNIII repeats. Several crystal structures were determined for N-terminal fragments including Ig1-4 from Sdk1 and Sdk2, Ig1-5 for Sdk2, and a chimera which includes the Ig1,2 from Sdk1 and Ig3,4 from Sdk2. All formed horseshoe-like structures with Ig1 and 2 aligned in a linear fashion followed by a short loop between Ig2 and Ig3, as the ectodomains flip back with alignment of Ig3 and Ig4 in parallel with Ig1 and Ig2. Contacts between Ig1 and Ig4 and between Ig2 and Ig3 stabilize the horseshoe. Remarkably, although other Ig superfamily homophilic adhesion molecules also form this characteristic horseshoe structure, the contacts between monomers underlying dimer formation are very different re-emphasizing how uncanny this three-dimensional structure is in generating different recognition strategies. A set of binding studies on mutant proteins provides strong support for the biological significance of the crystal structures. All of the experiments are well designed and executed to yield clear answers to a series of logically constructed questions. The combined data allow the authors to propose an insightful model for the mechanism of action of Sdk1 and Sdk2 which will be of substantial interest to neurobiologists and structural biologists alike. The hypothesis that the strength of the homophilic cis interactions serves to deter heterophilic, and hence favour homophilic, trans interactions is salient. In sum, the work is a strong candidate for *eLife*.

Essential revisions:

1) Both for Sdk1 and Sdk2, the homo-dimer affinity of Ig1-6 is about 5 times stronger than Ig1-4. However, the data presented in the manuscript argue against any roles of Ig5-6 in both molecules. The 5-fold affinity difference is likely to be significant. The authors need to address/discuss this point in their revision.

2) The data presented in Figure 6 address the origin of the specificity of homophilic vs. heterophilic interactions. The data is good. However, the interpretation can be argued. The authors argue that the L29/E168 interaction is favored in Sdk1 and the M29/D168 interaction is favored in Sdk2. It seems that both pairs of the interactions are not that favorable. It is just that the swapped heterophilic interactions are more unfavorable. While the final outcome is more or less the same, but the interpretation may be confusing to readers from outside the structural biology field.

3) Related to the point above, the authors present evidence that two amino acids play a role in regulating the specific of Sdk1 and Sdk2 preferential homophilic binding. I think these studies could be further explored. As there are only 5 interfacial residues different at the Ig1-2 interface, it would be interesting to assess whether changing all would lead to clean swap in specificity. It will be preferred if the authors have additional data on this. If not, the authors may want to discuss this point. Another possibility is to adjust the text in the sections dealing with homophilic vs. heterophilic interactions by covering alternative interpretations/possible weaknesses for the data available.

4) Figure 5: Although it is clear that the clustering of WT Sdk1 and N22R Sdk1 have difference, but the difference is not as great as the authors have stated. There are clear clustering puncta for the mutant (in the cell peripheries in particular). The authors perhaps should use line-scanning to quantify the distribution differences. Also for Figure 5, the figure looks convincing, whether the authors have a way of comparing cell surface expression levels for wt and mutant. It could be argued that very different expression levels could lead to differing outcomes? In Figure 5, the authors should clarify whether the puncta are surface features, some comment on this could be reassuring? Finally, the figure legend does not seem to match the figure presented. The panel also missing a scale bar.

5) The authors use clustering of Sdks on isolated cells as a read out for cis dimer formation. The clusters, of course, are much larger aggregates and as such the dimers are necessary to form higher order structures. What parts of the molecule promote this interaction? For instance, Sdks may well form cis dimers independent of the horseshoe structure and oligomerization may then occur through interactions between horseshoes. Alternatively, cis interactions between the horseshoes may induce structural alterations which then allow for cis interactions between dimers through different domains. Some discussions on this point would add value to the paper.

---

## [Author Response]

*1) Both for Sdk1 and Sdk2, the homo-dimer affinity of Ig1-6 is about 5 times stronger than Ig1-4. However, the data presented in the manuscript argue against any roles of Ig5-6 in both molecules. The 5-fold affinity difference is likely to be significant. The authors need to address/discuss this point in their revision.*

As requested, we now discuss this issue in the paper. There is precedent for the effects on binding affinity of domains that do not interact directly. Specifically, we cite the example of human VE-cadherin, in which EC1–5 has a ~4-fold greater dimerization affinity than EC1–2 (1.03 vs. 4.38 µM), even though the entire dimerization interface is contained within EC1 (Brasch et al., 2011 doi:10.1016/j.jmb.2011.01.031). We speculate that the effect may be due to entropic loss in the unbound state due to greater non-specific steric interactions. This in turn would drive the dimerization equilibrium towards the bound state.

*2) The data presented in Figure 6 address the origin of the specificity of homophilic vs. heterophilic interactions. The data is good. However, the interpretation can be argued. The authors argue that the L29/E168 interaction is favored in Sdk1 and the M29/D168 interaction is favored in Sdk2. It seems that both pairs of the interactions are not that favorable. It is just that the swapped heterophilic interactions are more unfavorable. While the final outcome is more or less the same, but the interpretation may be confusing to readers from outside the structural biology field.*

We agree with the Reviewers, and have now clarified the language describing the results of Figure 6, following the Reviewers’ suggestion.

*3) Related to the point above, the authors present evidence that two amino acids play a role in regulating the specific of Sdk1 and Sdk2 preferential homophilic binding. I think these studies could be further explored. As there are only 5 interfacial residues different at the Ig1-2 interface, it would be interesting to assess whether changing all would lead to clean swap in specificity. It will be preferred if the authors have additional data on this. If not, the authors may want to discuss this point. Another possibility is to adjust the text in the sections dealing with homophilic vs. heterophilic interactions by covering alternative interpretations/possible weaknesses for the data available.*

In response to the Reviewers’ comments we have extended our discussion of these data in the manuscript, noting the limitations with the current data in fully defining the basis of Sdk1 vs. Sdk2 specificity. Although further mutagenesis might be illuminating, it would require several months for a complete analysis. This is because we know from our previous work that teasing out individual contributions to affinity can be quite challenging. We have now discussed alternative possibilities and provided a more precise description of the role of the two amino acids in question.

*4) Figure 5: Although it is clear that the clustering of WT Sdk1 and N22R Sdk1 have difference, but the difference is not as great as the authors have stated. There are clear clustering puncta for the mutant (in the cell peripheries in particular). The authors perhaps should use line-scanning to quantify the distribution differences. Also for Figure 5, the figure looks convincing, whether the authors have a way of comparing cell surface expression levels for wt and mutant. It could be argued that very different expression levels could lead to differing outcomes? In Figure 5, the authors should clarify whether the puncta are surface features, some comment on this could be reassuring? Finally, the figure legend does not seem to match the figure presented. The panel also missing a scale bar.*

We have conducted additional experiments to address these points, and have added a new supplementary figure, Figure 5—figure supplement 2, to illustrate our results. Part A shows that wild type Sdk1 forms *cis*-clusters in isolated L-cells, independent of expression level, while the mutant only shows much smaller clusters. Part B presents line scans of immunofluorescence in these cells requested by the Reviewers, which show the distinct nature of wild-type and mutant localization patterns. Part C quantifies the difference in cluster size between wild-type and mutants. Part D shows cells immunostained live, confirming that the clusters are localized to the cell surface. Part E shows results of an ELISA-type assay performed on live-stained cells to demonstrate that similar levels of wild-type and mutant Sdk1 are present on the cell surface. Finally, Part F shows cells cotransfected with wild-type and mutant Sdk1, fused to YFP and RFP, respectively. These cells provide perhaps the most convincing evidence that the two proteins are differentially distributed within a single cell. (A small population occasionally colocalizes at the cell peripheries.) We have noted these new experiments in the main text.

We agree with the Reviewers that the mutant Sdk1 does sometimes aggregate at cell peripheries. We have no explanation, but speculate that the distinct cytoskeletal and adhesive structures present in these regions are responsible. We have therefore focused our attention on the central regions of cells, in which wild type Sdk1 forms clusters but mutant Sdk1 does not.

We also thank the Reviewers for noting the mistake in the legend to Figure 5, which we have now corrected.

*5) The authors use clustering of Sdks on isolated cells as a read out for cis dimer formation. The clusters, of course, are much larger aggregates and as such the dimers are necessary to form higher order structures. What parts of the molecule promote this interaction? For instance, Sdks may well form cis dimers independent of the horseshoe structure and oligomerization may then occur through interactions between horseshoes. Alternatively, cis interactions between the horseshoes may induce structural alterations which then allow for cis interactions between dimers through different domains. Some discussions on this point would add value to the paper.*

The large cell surface *cis*-clusters we observe in isolated cells are dependent on the Sdk-horseshoe dimer observed crystallographically, since disruption of the dimer interface with the N22R mutation results in a loss of *cis*-clustering. Given that a single *cis*-dimer is small, this interaction alone would be insufficient to underlie the large *cis* clusters observed. Therefore, we have inferred the existence of other interaction(s). However, the localization of interacting regions remains unknown. Until more data is obtained, we would prefer not to speculate further on the nature of this putative interaction, and its possible roles in Sdk function.